# A dissymmetric [Gd$_2$] coordination molecular dimer hosting six addressable spin qubits

Fernando Luis [1,2✉], Pablo J. Alonso[1,2], Olivier Roubeau[1,2], Verónica Velasco[3], David Zueco[1,2,4], David Aguilà[3], Jesús I. Martínez[1,2], Leoní A. Barrios[3] & Guillem Aromí[3✉]

Artificial magnetic molecules can host several spin qubits, which could then implement small-scale algorithms. In order to become of practical use, such molecular spin processors need to increase the available computational space and warrant universal operations. Here, we design, synthesize and fully characterize dissymetric molecular dimers hosting either one or two Gadolinium(III) ions. The strong sensitivity of Gadolinium magnetic anisotropy to its local coordination gives rise to different zero-field splittings at each metal site. As a result, the [LaGd] and [GdLu] complexes provide realizations of distinct spin qudits with eight unequally spaced levels. In the [Gd$_2$] dimer, these properties are combined with a Gd-Gd magnetic interaction, sufficiently strong to lift all level degeneracies, yet sufficiently weak to keep all levels within an experimentally accessible energy window. The spin Hamiltonian of this dimer allows a complete set of operations to act as a 64-dimensional all-electron spin qudit, or, equivalently, as six addressable qubits. Electron paramagnetic resonance experiments show that resonant transitions between different spin states can be coherently controlled, with coherence times $T_M$ of the order of 1μs limited by hyperfine interactions. Coordination complexes with embedded quantum functionalities are promising building blocks for quantum computation and simulation hybrid platforms.

[1] Instituto de Ciencia de Materiales de Aragón (ICMA), CSIC and Universidad de Zaragoza, Plaza San Francisco s/n, 50009 Zaragoza, Spain. [2] Dpto. de Física de la Materia Condensada, Universidad de Zaragoza, Pedro Cerbuna 12, 50009 Zaragoza, Spain. [3] Departament de Química Inorgànica and IN2UB, Universitat de Barcelona, Diagonal 645, 08028 Barcelona, Spain. [4] Fundación ARAID, Av. de Ranillas 1-D, 50018 Zaragoza, Spain. ✉email: fluis@unizar.es; guillem.aromi@qi.ub.edu

Electronic spins in solids are natural candidates to encode qubits, the basic units of future quantum computers[1]. Quantized spin projections give rise to a discrete set of states that can be coherently manipulated by external magnetic field pulses. In addition, they are insensitive to electric noise and can therefore show longer coherence times than qubits based on electric (e.g. superconducting) circuits. Outstanding examples are magnetic defects in semiconductor hosts, such as $NV^-$ centres in diamond[2] or $P^+$ donors in silicon[3]. These systems are however not easy to tune, and wiring them up into a scalable architecture remains very challenging. A promising alternative to address these questions relies on the use of electron spins hosted by artificial molecules, i.e., organic radicals or transition-metal complexes[4–6]. This chemistry-based approach affords the synthesis of macroscopic numbers of identical qubits. The design of the metal coordination provides some control over the spin levels, which determine the qubit frequencies[7,8]. Furthermore, the spin coherence can be enhanced by either reducing the number of neighbouring nuclear spins or by engineering qubit states that are insensitive to magnetic field fluctuations. As a result, molecular spin qubit coherence times $T_M$ have shown very significant improvements over the past years[8–13].

But arguably the most appealing trait of this approach is that it offers the possibility of scaling up computational resources within each molecular unit. The most straightforward strategy is to incorporate several (two or more) magnetic centres, each of them realizing a qubit. Several spin carriers can be assembled in a rational way, such that their nature, disposition and interaction result in a set of addressable transitions that allow coherently manipulating all qubit states[6]. At the synthetic level, this implies a necessary dissymmetry between the different magnetic centres. Early successful examples in either homometallic[14,15] or heterometallic[16,17] lanthanide dimers have shown that conditions to realize both CNOT and √SWAP two-qubit gates can be fulfilled[18], and allowed measuring the quantum coherence of a CNOT gate with $T_M$ of about 400 ns[19–21]. Also, modular supramolecular approaches able to incorporate multiple weakly coupled $Cr_7Ni$ qubits have been developed[22] and used to implement C-PHASE two-qubit gates[23]. A second strategy is to profit from the internal spin levels of each magnetic centre to create $d$-dimensional ($d > 2$) quantum systems, that is, qudits. Examples include the coherent manipulation of the multiple nuclear spin states of $^{159}TbPc_2$ ($I = 3/2$, $d = 4$)[24,25], $Et_4N[^{163}DyPc_2]$ ($I = 5/2$, $d = 6$)[26] and $^{173}Yb$(trensal) ($I = 5/2$, $d = 6$)[27] mononuclear complexes. The former of these enabled the first realization of a three-level Grover quantum search algorithm in a single molecule, using tools of molecular electronics to read-out the qudit state[28]. Electronic spin qudits can also be realized in some cases, for instance by using the $S = 7/2$ spin manifold of a Gd(III) ion provided that it has a sufficiently low magnetic anisotropy to make all levels experimentally accessible[29].

The integration of multiple levels in well-defined complexes increases the density of quantum information handled by these systems and could allow embedding specific functionalities, say quantum error correction or some simple quantum algorithms and simulations[27,30–32], at the molecular scale. These applications demand increasing the dimension of the computational space beyond three qubits while retaining the ability to perform universal quantum operations[30,33]. Meeting both conditions in a molecule still represents a daunting challenge.

Here we show that it is possible to design and synthesize such building blocks via the combination of the two strategies outlined above. The rational design of a coordination complex with multiple and magnetically inequivalent Gd(III) ions would allow to scale up to six or more addressable qubits within one single molecule, with the potential of realizing different quantum gate operations. We focus here on the realization of this scaling-up in a dissymmetric [Gd₂] molecule in which the two Gd(III) ions have a slightly different coordination[14,15]. The isolated properties of each Gd(III) $d = 8$ qudit are first determined through the isolation and study of the [LaGd] and [GdLu] analogue molecules, in which the Gd(III) ions occupy either of the two coordination sites present in [Gd₂]. The results show that the [Gd₂] molecule holds the promise to act as a six-qubit quantum processor or as a $d = 64$ electronic spin qudit.

## Results

**Synthesis and structures.** We previously showed that the ligand $H_3L$ forms a series of dissymmetric homometallic dilanthanide complexes of formula $(Hpy)[Ln_2(HL)_3(NO_3)(py)(H_2O)]$, where $H_3L$ = 6-(3-oxo-3-(2-hydroxyphenyl)propionyl)-pyridine-2-carboxylic acid and py = pyridine, among which the [Gd₂] compound studied here[14,15]. The two different Ln environments resulting from the different coordination pockets created by the three $HL^{2-}$ ligands ($HL^{2-}$ = double deprotonated $H_3L$) and their disposition give rise to two metal sites with markedly different Ln–O and Ln–N bond lengths, one position being systematically larger than the other. This allows forming in a controlled manner heterometallic dilanthanide molecules, provided their ionic radii are sufficiently different[16,17,34]. This unique synthetic strategy was first used to isolate the [CeEr], [LaEr] and [CeY] analogues[19]. Here, in order to study each Gd(III) ion present in [Gd₂], albeit isolated from the other, the [LaGd] and [GdLu] analogues are prepared in a similar manner. Note that the differences in ionic radii $\Delta r_i$, respectively of 9.4 and 7.7 pm[34], are sufficient to expect the formation of homogeneous heterometallic single phases in both cases[16,17]. Thus, equimolar amounts of $Gd(NO_3)_3$ and either $La(NO_3)_3$ or $Lu(NO_3)_3$ were made to react with $H_3L$ in pyridine. The layering of the resulting clear yellow solutions with $Et_2O$ resulted in homogeneous phases of yellow crystals that were found by single-crystal X-ray diffraction to consist of $(Hpy)[LaGd(HL)_3(NO_3)(py)(H_2O)]\cdot5py$ ([LaGd]) and (Hpy) $[GdLu(HL)_3(NO_3)(py)(H_2O)]\cdot5py$ ([GdLu]).

Compounds [LaGd] and [GdLu] crystallize in the monoclinic space group $P2_1/n$, with the asymmetric unit coinciding with their formula and $Z = 4$ (see Supplementary Table 1 and the CIF files available as Supplementary Data 1, for [LaGd], and Supplementary Data 2, for [GdLu]). The pyridinium cation forms a hydrogen bond with one of the carboxylic oxygen atoms from one of the $HL^{2-}$ ligands. Two of the five lattice pyridine molecules are hydrogen bonded to the coordinated water molecule (Supplementary Fig. 1, Supplementary Table 2). The metal complexes (Fig. 1) show the structure observed consistently within this series of compounds, with two lanthanide ions bridged and chelated by three $HL^{2-}$ ligands in two opposite orientations with respect to the La···Gd or Gd···Lu vector. Thus two different coordination environments are present, with two/one (O,N,O) pockets and one/two (O,O) chelates respectively for site 1/2. The coordination sphere is completed by a nitrate ion on site 1 (La in [LaGd] and Gd in [GdLu]), and by one pyridine molecule and a water molecule on site 2 (Gd in [LaGd] and Lu in [GdLu]). In [LaGd], the coordinated nitrate is bidentate, resulting in a coordination number CN of 10 for the largest La(III) ion, while the rest have CN of 9. The intermetallic separations are respectively 3.8941(3) and 3.761(1) Å for La···Gd and Gd···Lu (Supplementary Table 3), to be compared with the intermediate 3.804(1) Å Gd···Gd separation in [Gd₂]. The average Ln–O bonds to $HL^{2-}$ donors are longer at site 1, 2.544 Å (La in [LaGd]) and 2.432 Å (Gd in [GdLu]), than at site 2, 2.405 Å (Gd in [LaGd]) and 2.342 Å (Lu in [GdLu]). The differences in average bond distances $\Delta O$ between both sites are thus respectively 0.139 and

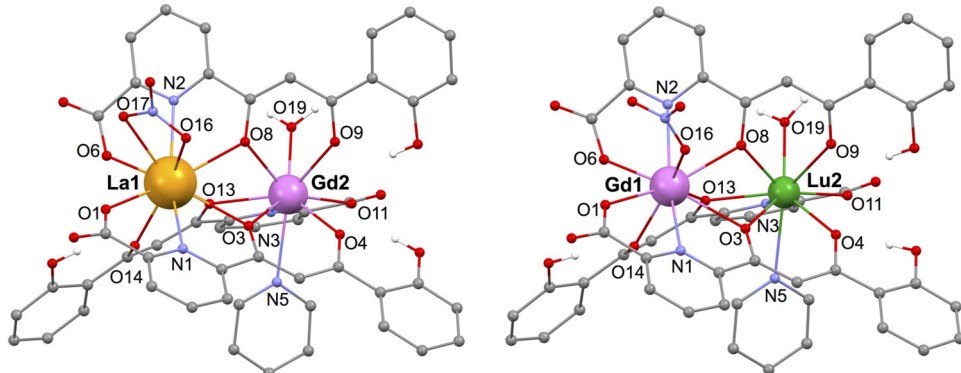

**Fig. 1 Molecular structures of [LaGd] and [GdLu] spin qudits.** For clarity, hydrogen atoms not involved in H-bonding and lattice pyridine molecules are omitted. Atomic sites involved in the coordination are labelled. Colour code: O, red; N, light blue; C, grey.

0.090 Å, larger than those observed in the homometallic analogues [La$_2$] ($\Delta O = 0.045$ Å) and [Gd$_2$] ($\Delta O = 0.026$ Å)[14], an indication that the lanthanide ions have taken selectively the optimal relative positions following their respective ionic radii[16,17].

The coordination sphere at both Gd sites also differs in shape, as indicated by continuous shape measures (Supplementary Fig. 2)[35,36]. Thus, the Gd2 site in [LaGd] is closest to an ideal capped antiprism ($C_{4v}$ symmetry) and to an ideal tricapped trigonal prism ($D_{3h}$ symmetry). The Gd1 site in [GdLu] is more irregular as indicated by distances to any ideal polyhedron 3–4 times larger. These differences are identical to those found between the two Gd sites in [Gd$_2$]. The metal site composition is confirmed for both compounds by the unreasonable relative displacement parameters and worse final refinement obtained for any other distribution of metals within the molecular structure.

The molecular metal composition observed in the solid-state is confirmed by Electrospray Ionization Mass Spectrometry (ESI-MS), which also allows ensuring the integrity of the molecular structure in solution. Thus, the spectra obtained from solutions of [LaGd], [Gd$_2$] and [GdLu] each exhibit a prominent peak with respectively $m/z$ of 1146.98, 1166.01 and 1183.02 and isotopic distribution that agree with the corresponding [LnLn'(H$_2$L)(HL)$_2$]$^+$ fragment. These fragments correspond to the molecular moiety in the absence of the terminal ligands, which are the only species detected by the technique of the dissociation equilibrium. In the case of [GdLu], a small impurity of [Gd$_2$] is observed indicating a process of partial scrambling. Indeed, the dissociation of the terminal ligands may conduce to a relaxation of the structure and therefore to a decrease of the selectivity as it was shown by previous ESI-MS and density functional theory studies with compounds of the same family[16,17]. Nevertheless, the selectivity remains important and follows the relative $\Delta r_i$, which for [LaGd] and [GdLu], is relatively high. While the ESI-MS technique does not allow a quantitative analysis, since the [Gd$_2$] impurity arises from the dissociated fraction (expected to be marginal) and the selectivity of the full molecule is that of the solid state, we are confident the amount of impurity in solution remains negligible. Therefore, the heterometallic molecular composition seen in the solid-state structure is maintained in solution, which is relevant for the studies of coherent spin dynamics done on diluted solutions. These studies confirm that any fraction of [Gd$_2$] in the solution of [GdLu] is not detectable since this would have an effect on the quantum coherence in the opposite direction as that observed (see below).

**Isolated qudits in [LaGd] and [GdLu]: magnetic dissymmetry.** A crucial requirement to properly define a qudit, or $N$ qubits,

using a system with multiple energy levels $d = 2^N$ is that the energy spectrum has some nonlinearity, i.e., that the levels are not simply equidistant as those of a harmonic oscillator[30]. In the case of Gd(III), with its ground state $S = 7/2$, this condition relies on the existence of a finite magnetic anisotropy[7,29]. An advantage of this ion is that, because of its $L = 0$ configuration, the intrinsic anisotropy of the free ion is negligible. Any zero-field splitting of the $d = 8$ spin levels necessarily arises from small distortions that the coordination environment induces on the close to spherical 4f electronic shell. This property makes Gd(III) a kind of model crystal field probe and allows modifying the magnetic anisotropy via changes in the local coordination. Often, this anisotropy is also quite small, much smaller than those typically found for other lanthanides with $L \neq 0$[4,5,37–39], thereby making these levels accessible via conventional magnetic spectroscopy techniques.

The continuous-wave electron paramagnetic resonance (cw-EPR) spectra of [LaGd] and [GdLu] are shown in Fig. 2. The experiments have been performed on powdered samples using both X-band (frequency $\omega/2\pi = 9.886$ GHz) and Q-band ($\omega/2\pi = 33.33$ GHz) spectrometers. Spectra measured on each sample at different frequencies do not simply scale with $H/\omega$, where $H$ is the external magnetic field. This shows already that Gd(III) acquires a net magnetic anisotropy in the two possible coordination sites 1 and 2 (Fig. 1). Besides, the spectra of the two samples are also different, thus showing that the magnetic anisotropy depends on the coordination environment (Fig. 2e).

In order to render these arguments quantitative, we have performed fits of the experimental spectra using the EPR simulation package EasySpin[40]. Taking into account the low symmetry of the Gd coordination sites in these molecules, we have considered the simplest lowest-order spin Hamiltonian

$$\mathcal{H} = -DS_z^2 + E\left(S_x^2 - S_y^2\right) - g\mu_B \mathbf{HS}, \qquad (1)$$

where $D$ and $E$ are zero-field splitting parameters and $g$ is the electron spin g-factor. The results of these fits are compared to the experimental results in Fig. 2. A reasonably good agreement was obtained by setting $D = 2$ GHz, $|E| = 0.67$ GHz and $g = 1.99$ for [LaGd] and $D = 2.4$ GHz, $|E| = 0.8$ GHz and $g = 1.99$ for [GdLu]. The fact that the orthorombicity $|E|/D$ is close to the maximum value (1/3) probably arises from the low symmetry of both coordination sites. The line broadenings suggest that there are sizeable distributions in $D$ and $E$, of about 60% for both compounds.

Another experimental technique that provides information on the structure of electronic energy levels is heat capacity. Results obtained for the two molecular "monomers" are shown in Fig. 3. Above 10 K, the specific heat $c_P/R$ is dominated by excitations of vibrational modes. This lattice contribution had been measured

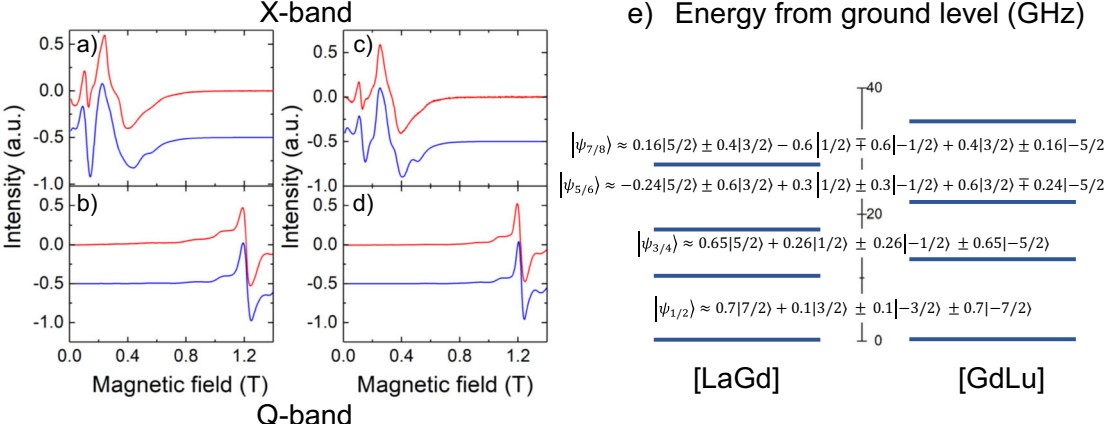

**Fig. 2 Magnetic spectroscopy of [LaGd] and [GdLu] spin qudits.** X-band and Q-band cw-EPR spectra measured on powdered samples of [LaGd] (**a**, **b**) and of [GdLu] (**c**, **d**) at $T = 6$ K. The red solid lines are experimental data, whereas the blue solid lines are simulations obtained with the spin Hamiltonian (1) and the parameters given in the text. The latter have been shifted down for clarity. Panel **e** compares the splitting of the Gd(III) spin levels at zero magnetic fields in both molecules, which correspond to the coordination sites 1 and 2 in the structures of Fig. 1. The wave functions of all level doublets are also indicated in this panel.

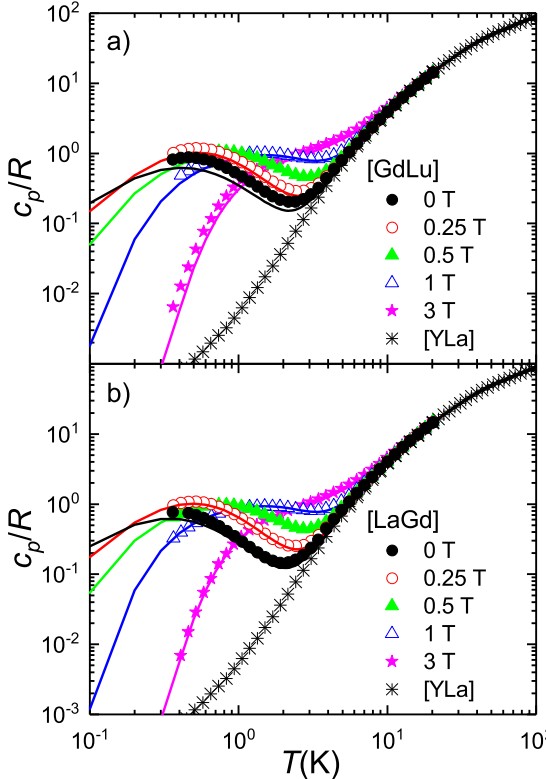

**Fig. 3 Field-dependent specific heat of [GdLu] and [LaGd] spin qudits.** Curves were measured at zero and different applied magnetic fields on powdered samples of [GdLu] (**a**) and [LaGd] (**b**). Data measured for the diamagnetic complex [YLa][19], which arise from vibrational excitations only, are also shown in both panels. The solid lines are numerical calculations of the magnetic heat capacity, derived from Eq. (1) with the same parameters, and their distributions, that account for the cw-EPR data of Fig. 2 to which the nonmagnetic contribution has been added.

directly on the diamagnetic derivative [YLa][19], and agrees very well with the high-temperature behaviour. The additional anomaly observed, at $H = 0$, for both [LaGd] and [GdLu] below 2 K must then be associated with the finite splitting of the Gd(III) electronic spin levels. A nice feature of these results is that the position of the specific heat maximum provides a direct measure of the overall zero-field splitting[41].

The comparison between results measured for [LaGd] and [GdLu] confirms that the magnetic anisotropy is slightly stronger for the latter. The Schottky-like broad maximum is indeed well accounted for by numerical calculations based on Eq. (1) using the same $D$ and $E$ parameters derived from EPR experiments (and with the same $D$- and $E$- strains). A good agreement is also found for data measured under non-zero magnetic fields. Results for the two compounds become progressively closer to each other as $H$ increases, due to the relatively weak magnetic anisotropy of Gd(III) and to the predominance of the Zeeman term in Eq. (1) for sufficiently strong $H$.

In conclusion, the results shown in this section demonstrate that Gd(III) ions coordinated in the molecular structures of [LaGd] and [GdLu] have low-lying electronic energy level schemes that provide a basis for two different spin qudits.

**Exchange coupling in [Gd₂].** In this section, we turn our attention to the molecular dimer [Gd₂]. This complex hosts the two magnetic ions, in different coordination sites, whose properties in isolation have been discussed in the previous section. This discussion suggests that this molecule, with $(2S + 1) \times (2S + 1) = 64 = 2^6$ unequally spaced levels, provides a proper implementation of a $d = 64$ qudit or of six qubits. However, an additional necessary ingredient, related to the condition of universality that we discuss below, is the existence of a net coupling between the two spins.

In order to get information of the spin-spin interaction within the molecule, we have compared the magnetic, spectroscopic and thermal properties of the [Gd₂] complex with those measured on [LaGd] and [GdLu]. Figure 4 shows the magnetic response of the three samples. While the $\chi T$ plots of [LaGd] and [GdLu] agree with the predictions for isolated Gd(III) ions (an almost temperature-independent value, in agreement with Curie's law), the data for [Gd₂] show a decrease below ~4 K. This behaviour is compatible with the existence of a weak isotropic coupling described by the following spin Hamiltonian

$$\mathcal{H} = \mathcal{H}_1 + \mathcal{H}_2 - J\mathbf{S}_1\mathbf{S}_2 \qquad (2)$$

where H₁ and H₂ are the spin Hamiltonians of each isolated Gd (III) ion, given by Eq. (1) with the appropriate parameters, and $J$ is the spin-spin coupling constant. As a further simplification, the

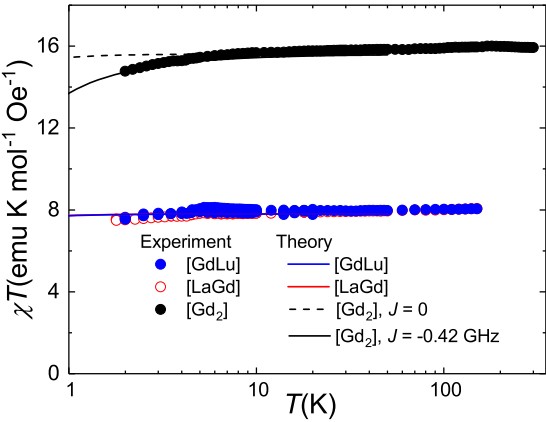

**Fig. 4 Magnetic coupling in the [Gd₂] dimer.** Product of the molar susceptibility χ times temperature measured on powder samples of [LaGd], [GdLu] and [Gd₂]. The lines are numerical calculations. In the case of the molecules hosting only one magnetic ion, they are derived from the spin Hamiltonian (1) with the parameters determined independently from EPR experiments (Fig. 2). In the case of [Gd₂], the calculations include also a spin-spin coupling, as in Eq. (2), with two values for the interaction constant $J$.

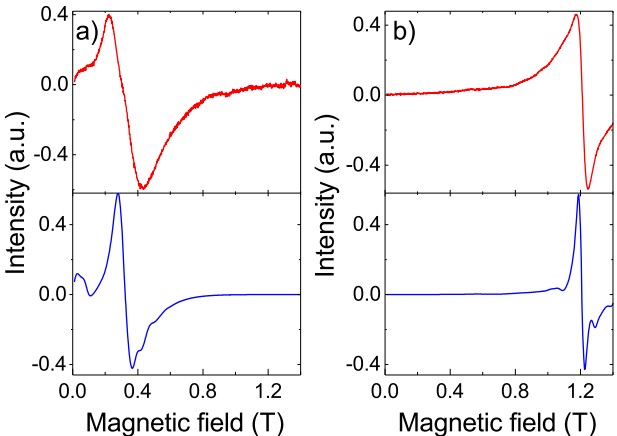

**Fig. 5 Magnetic spectroscopy of the [Gd₂] dimer.** X-band (**a**) and Q-band (**b**) cw-EPR spectra measured on a powder sample of [Gd₂] at $T = 6$ K. The curves at the top are experimental results, whereas those at the bottom are results of numerical calculations derived from Eq. (2) with $J = -0.42$ GHz.

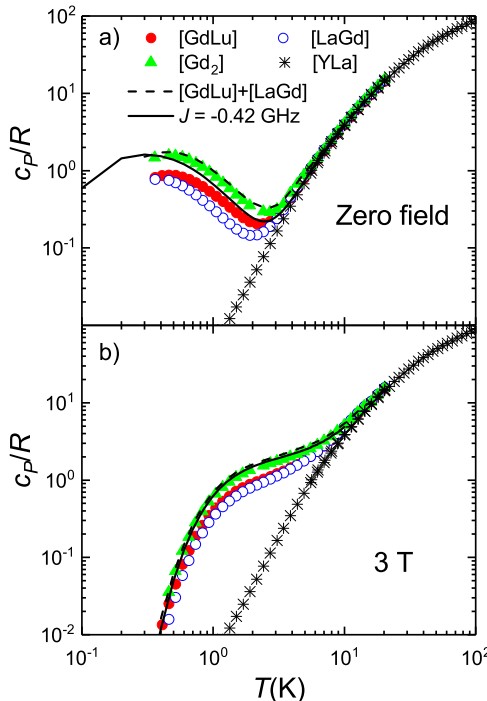

**Fig. 6 Field-dependent specific heat of the [Gd₂] dimer.** Specific heat of [Gd₂], compared to those of monomers [LaGd] and [GdLu] and of the diamagnetic [YLa][19], measured at zero magnetic fields (**a**) and under a strong 3 T magnetic field (**b**). The dashed line represents the case in which the two spins in [Gd₂] would not interact. It was obtained by adding the results measured on [LaGd] and [GdLu]. The solid line shows the results of a numerical calculation based on Eq. (2) with $J = -0.42$ GHz.

anisotropy axes at sites 1 and 2 have been taken as collinear. A reasonably good agreement with the experimental results is found for an antiferromagnetic interaction with $J = -0.42$ GHz (see Fig. 4). This interaction is likely dominated by intramolecular dipole-dipole interactions, which would give rise to $-0.92$ GHz $\leq J \leq +2.73$ GHz, depending on the orientation of the main anisotropy axis $z$. Therefore, Eq. (2) must be regarded as a simplified description, with an effective isotropic $J$, of the energy level scheme and the overall magnetic moment of [Gd₂].

The results of EPR measurements performed on [Gd₂] are also compatible with the existence of magnetic interaction between the two Gd ions. Figure 5 shows the results measured at X-band and Q-band. These spectra are not simple superpositions of those measured on the [LaGd] and [GdLu] monomers (see Figure S3). The strength of $J$ is, however, quite small as compared with that of the single-ion anisotropies. Whereas the latter give rise to zero-field splittings of order 20–30 GHz (Fig. 2) the energy scale of the $J\mathbf{S_1S_2}$ term is about 4 GHz. For this reason, it is not possible to accurately determine $J$ solely from EPR data. Yet, as Fig. 5 shows,

the results are compatible with calculations performed with the same $J$ inferred from magnetic measurements, albeit with an additional broadening that might point to the existence of some 'J-strain' or, as it might be expected from the discussion above, some anisotropy in the coupling between the two Gd spins.

Similar considerations apply to the results of heat capacity experiments, which are shown in Fig. 6. The Schottky anomaly associated with the magnetic anisotropy of both ions dominate data measured above 0.35 K. Still, these data are compatible, within the uncertainties of the experiment and the underlying model, with the predictions derived from Eq. (2) for $J = -0.42$ GHz. For sufficiently strong magnetic fields (see Fig. 6b), the differences between the specific heat data of isolated and coupled spins (and even between those of [LaGd] and [GdLu]) tend to vanish, as expected.

**Magnetic energy level scheme: six-qubit encoding and universal operation.** The results described in previous sections show that Eq. (2) provides a reasonably good account of the low-lying magnetic energy levels of [Gd₂] and of all experimental quantities that derive from it. We next discuss, on the basis of this description, the potential of this molecular dimer to implement multiple qubits. The energy scheme of this molecule in a magnetic field, shown in Fig. 7a, consists of a set of 64 unequally spaced levels, with level separations between adjacent levels of a few GHz. This scheme obviously admits a labelling of the levels in terms of the states of a qudit (from $|n = 1\rangle$ for the ground level to $|n = 64\rangle$ for the highest excited one) or in terms of the states of six qubits (say, from $|000000\rangle$ to $|111111\rangle$). However, this condition, i.e., that the space dimension is large enough, is not sufficient to ensure that such a small "processor" could perform universal quantum operations. The condition of universality

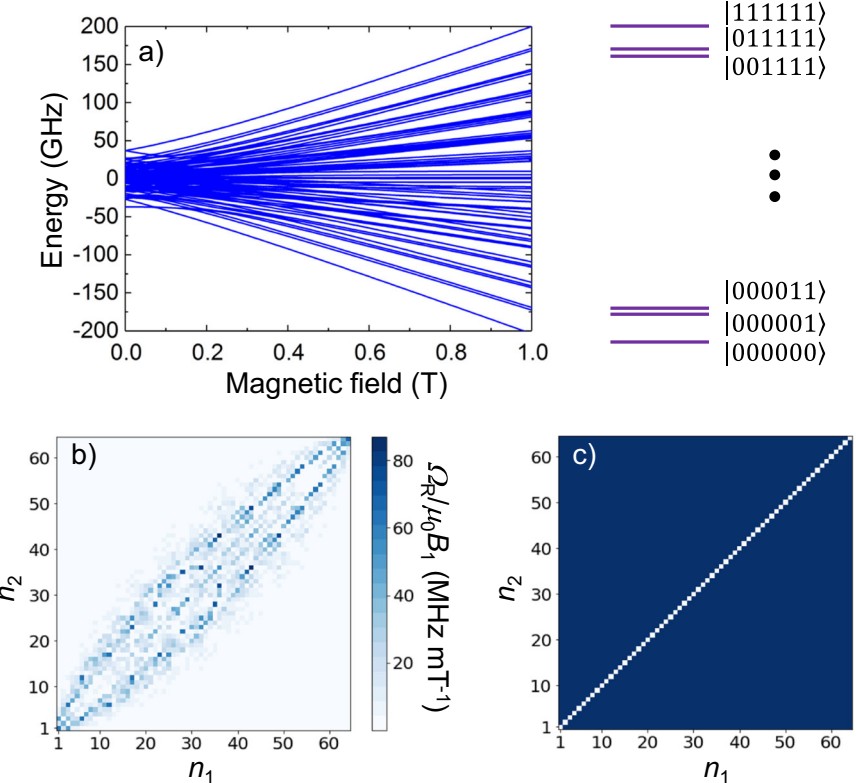

**Fig. 7 Six-qubit encoding and universality in the [Gd$_2$] dimer. a** Energy level scheme of [Gd$_2$] calculated with Eq. (2) and the parameters given in the text. The magnetic field is oriented along the diagonal between the magnetic axes $x$, $y$ and $z$ of the two Gd(III) ions, which are taken as collinear for simplicity. Possible labelling of the levels in terms of the basis states of six qubits is shown, in which adjacent levels differ only in the state of one qubit. **b** Colour map of the Rabi frequencies, calculated at 0.5 T, linking different levels, numbered from 1, for the ground level, up to 64, for the highest energy one. **c** Map of transitions accessible by concatenating different allowed resonant transitions at the same field, showing that the [Gd$_2$] system allows universal operations.

implies that any quantum superposition of the basis states (the 64 mentioned above) can be generated starting from any initial state, e.g., starting from the system initialized in its ground state. It implies that there exist non-forbidden transitions between different levels, which can be univocally addressed by setting the frequency of a microwave pulse or the external magnetic field, and which form a complete set, in the sense, defined above, of allowing "visiting" all possible spin states.

Figure 7b shows the Rabi frequencies for photon-induced resonant transitions between any two levels. This plot shows a dense map of allowed transitions, with rates exceeding 10–15 MHz mT$^{-1}$, which link the ground state with any other basis state. In order to perform a more rigorous demonstration, we have adapted the mathematical proof for universality[42], based on the formalism of Lie algebras, to this particular situation. Figure 7c shows that any two states of the basis can be connected by concatenation of resonant transitions. This question has received relatively little attention in connection with molecular qubits, but it is not as simple to achieve as it might appear at first sight. Its relationship with the formal structure of the underlying spin Hamiltonian can be better understood by comparing the situation of the two constituent Gd ions, realized independently in [LaGd] and [GdLu] complexes, with that of [Gd$_2$]. Applying the same mathematical method, we find that the Hamiltonian given by Eq. (1) affords a complete set of operations for each qudit (see Supplementary Fig. 4). However, Eq. (2), which describes the two qudits in [Gd$_2$], is universal only when $J \neq 0$ (compare Fig. 7 with Supplementary Fig. 5, which illustrates the situation for $J = 0$). The reason for this condition, which we anticipated above, is that conditional operations between states of

the two Gd(III) ions can be implemented with simple resonant pulses if and only if each of them has some anisotropy and the two are magnetically coupled.

**Quantum spin coherence and relaxation.** The previous section shows that [Gd$_2$] provides a suitable platform for a six-qubit quantum processor, in the sense of having the right Hilbert space and a sufficient number of allowed transitions. However, it also sets a final stringent condition, namely that transitions between these levels can be implemented coherently. The latter depends on the spin-coherence and spin-relaxation times, which we discuss in this section. The spin dynamics has been experimentally studied, at 6 K, on diluted solutions of [LaGd], [GdLu] and [Gd$_2$] in MeOH-$d^4$:EtOH-$d^6$, with concentrations in the range 0.38–0.61 mmol L$^{-1}$. As is well known[9], the use of deuterated solvents reduces the hyperfine couplings with the solvent nuclear spins and enhances the electronic spin coherence. The comparison with results measured for a sample of [GdLu] diluted in a non-deuterated solvent mixture, shown in Supplementary Fig. 6a, confirms this enhancement. For all three molecular systems, Electron Spin-Echo (ESE) signals are observed over the entire 10–810 mT range of magnetic fields used, either using 2- or 3-pulse sequences (Supplementary Figs. 6–9).

Therefore, both the isolated qudits in [LaGd] and [GdLu] and the exchange-coupled pair in [Gd$_2$] present measurable quantum coherence over the full magnetic field range, thus supporting the picture discussed in the previous section. The echo intensity varies with $H$, which allows obtaining echo-induced EPR spectra. These are in good agreement with the cw spectra, although the presence of a strong modulation of the echo decay (see below)

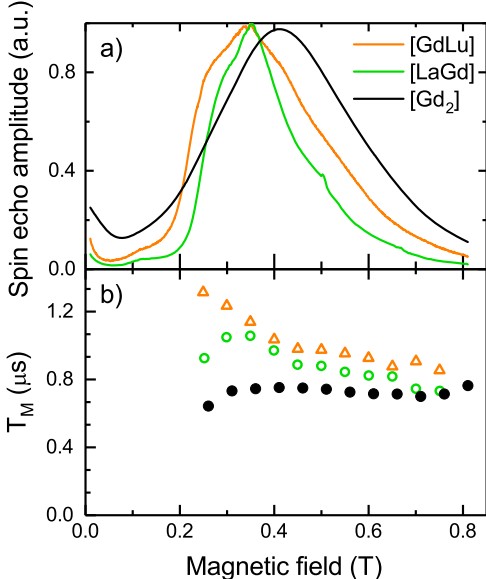

**Fig. 8 Spin coherence of the molecular qudits.** Comparison of the ESE-detected EPR spectra (**a**) and the field dependence of the phase memory times $T_M$ (**b**), measured on diluted MeOH-$d^4$:EtOH-$d^6$ solutions of [LaGd], [GdLu] and [Gd$_2$] at 6 K.

results in variations with the time interval between pulses (Supplementary Figs. 7 and 8).

Phase memory times $T_M$ have been obtained by measuring the spin-echo decay following a Hahn sequence of $\pi/2$ and $\pi$ pulses (typically 16 ns and 32 ns long, respectively) separated by a varying interval $\tau$. For all three compounds, the ESE intensity decreases in a similar manner with increasing $\tau$, the decay being slightly more rapid in the case of [Gd$_2$]. The ESE decay exhibits a strong modulation with a frequency that increases with $H$, independently of the compound. To quantify these effects, the ESE decays were modelled (Supplementary Fig. 10) through the equation

$$y(\tau) = y_0 + A_{2p}e^{-2\tau/T_M}\{1 + ke^{-\lambda\tau}\cos(2\pi\nu\tau + \phi)\}, \quad (3)$$

in which $y_0$ is a constant background, accounting for the floating level of the electronic output signal, $A_{2p}$ is the initial amplitude, $k$ the relative amplitude of the modulated signal, $\nu$ the average frequency of the oscillating component and $\lambda$ its decay rate, and $\phi$ the non-zero phase due to the detector dead-time. The magnetic field dependence of $A_{2p}$ exhibits a maximum in the range 300–450 mT, slightly broader for [GdLu] than [GdLa] (Supplementary Fig. 12). The [Gd$_2$] data are further broadened and cannot be obtained as a combination of those of [LaGd] and [GdLu], in agreement with the cw-EPR results (see Fig. 5 and Supplementary Figs. 3 and 9). The phase coherence times of the isolated qubits in [LaGd] and [GdLu] are similar, although slightly larger for the later (Fig. 8). In both cases, $T_M$ decreases with $H$, respectively from 1.05/1.31 μs at 300 mT to 0.73/0.85 μs at 750 mT. On the contrary, $T_M$ of [Gd$_2$] remains virtually constant at *ca.* 0.73 μs for all applied fields. Remarkably, the phase memory times derived for the isolated qubits in [LaGd] and [GdLu] are about twice longer than those found in a 1% magnetically diluted crystal of the polyoxometallate [GdW$_{30}$][29]. The significantly shorter coherent times found for [Gd$_2$] are in agreement with it being more sensitive to decoherence. This effect can be tentatively associated with the coupling between the two spins, which results in the presence of more possible excitations contributing to decoherence. Within this picture, states of the two coupled Gd (III) ions feel decoherence sources affecting either site 1 or site 2.

As it happens with other properties (see e.g. Fig. 6), the differences between $T_M$ of the monomers and of [Gd$_2$] are also progressively reduced as the magnetic field increases, likely because the Zeeman interaction begins then to dominate over the spin–spin coupling.

Spin-lattice relaxation times $T_1$ have been determined through spin-recovery measurements after a $\pi - T - \pi/2 - \tau - \pi$ pulse sequence, with a varying length of the first interval $T$ and a fixed $\tau$. In all three compounds, the intensity increases with $T$. These curves were modelled (Supplementary Fig. 11) with a single exponential function to determine $T_1$. The field-dependences of the inversion recovery amplitude $A_{IR}$ (Supplementary Fig. 13) resemble those of $A_{2p}$, again with broader maxima for [Gd$_2$], that cannot be obtained as a combination those of [LaGd] and [GdLu]. Values of $T_1$ for [LaGd] and [GdLu] lie in the range 2.5–3.5 μs (Supplementary Fig. 13) and depend only weakly on $H$. As for $T_M$, values of $T_1$ for the exchange-coupled [Gd$_2$] are significantly lower, *ca.* 1.5–2 μs.

These experiments give also a hint on possible sources of decoherence. The frequency of the ESE modulation in the 2-pulse experiments is very close to the Larmor frequency of the ²H nucleus over the whole range of magnetic fields and for the three compounds (Supplementary Fig. 12). The modulation can therefore be ascribed to the interaction with the "remote" deuterons of the solvents. The coupling to each solvent nucleus is very weak, but the sum of many contributions gives rise to the large modulation observed. The origin of the modulation was confirmed by measuring a solution of [GdLu] in non-deuterated solvents (Supplementary Fig. 6a) for which the frequency modulation corresponds to the larger Larmor frequency of ¹H. This suggests that in these experiments hyperfine interactions with the solvent are still very relevant and, therefore, that the maximally attainable $T_M$ for isolated molecules can be even higher than the values shown in Fig. 8.

**Coherent control: Rabi oscillations.** Spin nutation experiments provide information on the Rabi frequencies, to be compared with $T_M$, and serve to illustrate the difficulties that experiments on frozen solutions face when trying to implement quantum gates in spin qudits. Experiments were performed at 6 K on the same diluted MeOH-$d^4$:EtOH-$d^6$ solutions of [LaGd], [GdLu] and [Gd$_2$] described above. They involve the measurement of the ESE generated by a variable duration pulse ($0.1 \le t_p \le 1.6$ μs) refocused by a $\pi$ pulse, with $\tau$ intervals ($100 \le \tau \le 200$ ns) between the two pulses and between the refocusing pulse and ESE detection. Representative results are shown in Fig. 9, for [Gd$_2$], and in the supplementary material (Supplementary Figs. 14–16), for [LaGd] and [GdLu]. In all cases, a strongly damped coherent oscillation is observed. The decay is much faster than what one would expect for a single coherent transition between states with decoherence times $T_M \approx 0.8$–1.2 μs, even if one considers possible inhomogeneities in the amplitude $B_1$ of the micro-wave field pulse[43,44]. The fast decay arises instead from the destructive interference of Rabi oscillations between different states, with also different Rabi frequencies, that the initial microwave pulse is able to excite in a sample of randomly oriented molecules. A Fourier transformation of the time-dependent signals shows indeed sizeable distributions of Rabi frequencies in the three samples. At 410 mT and an attenuation of 10 dB, which corresponds to $B_1 \le 0.275$ mT[29], we find average $\Omega_R \approx 12$ MHz for [LaGd], $\Omega_R \approx 17$ MHz for [GdLu] and $\Omega_R \approx 20$ MHz for [Gd$_2$], which lie within the range of values expected for these systems (see Fig. 7 and Supplementary Fig. 4).

In addition, these Fourier plots reveal also two additional oscillations. Their frequencies are independent of $B_1$, unlike $\Omega_R$,

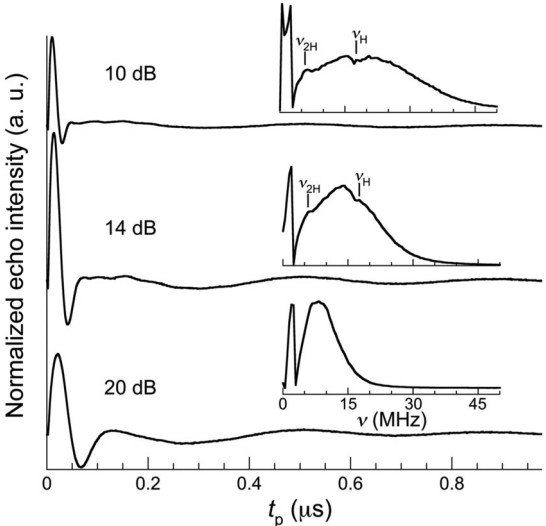

**Fig. 9 Spin nutation for the [Gd₂] dimer.** Experiments were performed at $T = 6$ K and for a 410 mT magnetic field, that is, close to the maximum observed in the echo-induced spectrum, and for three different microwave pulse powers (attenuations of 10, 14 and 20 dB corresponding to $B_1 \approx$ 0.275 mT, 0.16 mT and 0.092 mT, respectively), on a diluted MeOH-$d^4$: EtOH-$d^6$ solution of [Gd₂]. For each set of data, the Fourier transform is shown as inset, evidencing oscillations with characteristic frequencies that correspond to the Larmor frequencies of ²H and ¹H, as indicated, in addition to the main Rabi frequency.

but proportional to the external magnetic field, and agree well with the Larmor frequencies of the ²H and ¹H nuclei (Fig. 9 and Supplementary Figs. 14–16). They likely result from a cross-polarization of these nuclei by the Gd spin, known as Hartmann-Hahn effect[45], and confirm that the Gd spin is coupled to the surrounding deuterium and proton nuclear spins, contributing to the quantum spin decoherence.

## Discussion

In this work, we have designed a molecular structure that can host two Gd(III) ions in distinct coordination environments. Combined with the intrinsic properties of this ion, in particular, its $S = 7/2$ and zero orbital moment, this design enables realizing two different spin qudits, each with $d = 8$ or, equivalently, three qubits. These qudits can be studied in isolation, in either [LaGd] and [GdLu] molecules in which they are accompanied by a dia-magnetic counterpart or can be both integrated in the same molecular unit. In the latter case, we have shown that the molecular dimer [Gd₂] meets all conditions needed to realize six qubits (or a $d = 64$ qudit).

The integration of a high number of addressable quantum states in well-defined molecular units represents one of the distinctive traits of the chemical approach to quantum technologies and can provide a number of competitive advantages in this field. Such molecules can locally act as tiny quantum processors themselves, the equivalent of NISQS (Noisy Intermediate Size Quantum Systems) in other schemes[46], thus implementing simple algorithms. Of special relevance would be the correction of errors, e.g. phase errors that are likely to be most disturbing in the case of spin qubits, at the molecular scale[27,30]. For this, it is not even necessary that the dimension of the Hilbert space defined by the available spin states be a power of two. These molecular NISQS can also serve as a suitable basis where to map quantum simulations of simple systems, such as small molecules[33,47]. An advantage of using a single physical unit to perform these simple computations is that it reduces the number of non-local

operations, which require switching on and off interactions between different parts of a quantum circuit, and that are often more sensitive to decoherence and, therefore, error[31,32].

The fulfilment of this potential defines also a series of important challenges. A prominent one is how to individually address each of the relevant transitions. This requires working on well-oriented molecules, to avoid the difficulties discussed above in connection with nutation experiments and decreasing significantly the linewidth of each resonance. Even though coherence times are reasonably long in these systems, the resonance lines show a relatively large inhomogeneous broadening, which mainly arises from the random orientation of molecules in frozen solutions and from the existence of some distributions in the parameters that define the spin Hamiltonian (exchange and anisotropy constants). These difficulties can possibly be overcome by synthesizing well-organized molecular frameworks, in which all molecules need to be oriented in the same manner and, at the same time, magnetically diluted or sufficiently far away from each other[48–50]. In the end, the ideal situation would be to explore the response of individual molecules, either via the application of single-molecule electronics[24,25,28] or by enhancing the current sensitivity of magnetic spectroscopic techniques[51–54].

A second challenge is how to move forward beyond the level of a few qubits. As it becomes clear by inspecting the energy level scheme of a $d = 64$ qudit, the resonant transitions that provide the basic set of operations suffer already from a "frequency crowding", which would seriously hinder addressing individual resonances for qudits of increasing dimension. Therefore, scaling beyond this point must proceed by linking different units via coherent mediators. Implementing such switchable couplings within molecules (e.g. via chemical linkers sensitive to some external stimuli, such as light) remains a challenging goal[22,55,56]. The alternative is to combine a chemical, bottom up, approach with the integration of functional molecules into solid-state circuits. A recent proposal for a scalable architecture considers the possibility of applying superconducting circuits, namely on-chip resonators, to control, read-out and communicate magnetic molecules[57,58]. A further advantage of these devices, and of dealing with electronic spins having sizeable energy splittings, is that they are compatible with working at very low temperatures, which are necessary to initialize the qudit to its ground state (its population becomes larger than 99.99% for $T < 0.1$ K and a magnetic field of 0.5 T). Within this scheme, medium size molecular qudits, able to embed some critical functionalities such as phase error correction, would provide very attractive building blocks to reach computational performances difficult to match by other platforms.

## Methods

**Synthesis**. **6-(3-oxo-3-(2-hydroxyphenyl)propionyl)pyridine-2-carboxylic acid (H₃L)**. The ligand H₃L was synthesized as described previously[14].

**(Hpy)[Gd₂(HL)₃(NO₃)(py)(H₂O)]·5py** ([Gd₂]). Compound [Gd₂] was synthesized as described previously[15]. Purity was checked by elemental analysis, mass spectrometry and multiple cell determinations on single crystals, including full structure determination.

**(Hpy)[LaGd(HL)₃(NO₃)(py)(H₂O)]·5py** ([LaGd]). A 10 mL pyridine solution of ligand H₃L (30.0 mg, 0.105 mmol) is added slowly to a 10 mL pyridine solution of La(NO₃)₃·6H₂O (15.2 mg, 0.035 mmol) and Gd(NO₃)₃·6H₂O (15.8 mg, 0.035 mmol). The mixture is stirred at room temperature for one hour, yielding a clear shiny yellow solution that is layered with diethylether. Crystals of [LaGd] form slowly and are recovered after one weak in a 63 % yield. ESI-MS: [LaGd (HL)₂(H₂L)]⁺ $m/z = 1146.98$. Elemental analysis found (calc.) for [LaGd(H₂L) (HL)₂(NO₃)(py)(H₂O)]·3H₂O: C 44.11 (44.16); H 2.77 (3.04); N 4.87 (5.15). IR (KBr, υ/cm⁻¹): 3410 mb, 1618s, 1583 s, 1560 m, 1528 s, 1463 m, 1400 s, 1384 s, 1298 m, 1239w, 1204w, 1148w, 1120w, 1059 m, 949 m, 891w, 760 m, 706 m, 663 m, 635w, 568w.

**(Hpy)[GdLu(HL)₃(NO₃)(py)(H₂O)]·5py** ([GdLu]). A 10 mL pyridine solution of ligand H₃L (30.0 mg, 0.105 mmol) is added slowly to a 10 mL pyridine solution of Gd(NO₃)₃·6H₂O (15.8 mg, 0.035 mmol) and Lu(NO₃)₃·6H₂O (16.4 mg,

0.035 mmol). The mixture is stirred at room temperature for one hour, yielding a clear yellow solution to which 200 μL conc. $HNO_3$ is added. After 10 more min stirring, the solution is layered with diethylether. Crystals of [GdLu] are recovered after 2 weeks in a 53% yield. ESI-MS: $[GdLu(HL)_2(H_2L)]^+$ $m/z = 1183.02$. Elemental analysis found (calc.) for $[GdLu(H_2L)(HL)_2(NO_3)(py)(H_2O)] \cdot 6.7H_2O$: C 40.33 (41.06); H 2.56 (3.34); N 5.50 (4.79). IR (KBr, $\upsilon/cm^{-1}$): 3422 mb, 1619s, 1585 s, 1560 m, 1528 s, 1465 m, 1401 s, 1384 s, 1325 m, 1299 m, 1239w, 1208w, 1147w, 1121w, 1058 m, 950 m, 892w, 756w, 708 m, 666w, 637w, 569w.

**Magnetic measurements**. Magnetic measurements were performed using a Quantum Design SQUID MPMS-XL magnetometer through the Physical Measurements unit of the Servicio de Apoyo a la Investigación-SAI, Universidad de Zaragoza. All data were corrected for the sample holders and grease contributions, determined empirically as well as for the intrinsic diamagnetism of the sample, estimated using Pascal constants.

**Heat capacity experiments**. Heat capacity data were measured, down to $T = 0.35$ K, with a commercial physical property measurement system (PPMS, Physical Measurements unit of the Servicio de Apoyo a la Investigación-SAI, Universidad de Zaragoza) that makes use of the relaxation method. The samples, in powder form, were pressed into pellets and placed onto the calorimeter on top of a thin layer of Apiezon N grease that fixes the sample and improves the thermal contact. The raw data were corrected from the known contributions arising from the empty calorimeter and the grease.

**Electron paramagnetic resonance experiments**. Continuous-wave (cw) EPR measurements were performed with a Bruker Biospin ELEXSYS E-580 spectrometer operating in the X-band and Q-band. Solid-state cw-EPR measurements were performed at RT on polycrystalline samples placed in quartz tubes.

In addition, pulsed time domain (TD) measurements were performed at X-band frequencies. In these experiments, the typical widths of the π/2 and π pulses were 16 and 32 ns, respectively. In order to avoid unwanted echoes, a 2-step (4-step) phase cycle was used in the 2-pulse (inversion recovery and three-pulse) experiments. The high power microwave excitation was obtained by using a TWT amplifier. A dielectric low Q cavity from Bruker was used as resonator. TD-EPR measurements were done on frozen solutions, for which gas-flow Helium cryostats were used. The polycrystalline solids were dissolved in a 1:1 mixture of fully deuterated methanol and ethanol. The use of deuterated solvents is intended to limit decoherence due to protons. The TD-EPR experiments were performed at 6 K on 0.61 ([LaGd]), 0.38 ([GdLu]) and 0.44 ([Gd₂]) mmol L⁻¹ solutions. Measurements on 0.13 and 0.08 mmol L⁻¹ solutions of [GdLu] were also performed to discard any variation of $T_1$ and $T_M$ in the range of concentrations used. Additional TD-EPR measurements were performed on a 0.38 mmol/L solution of [GdLu] in a 1:1 mixture of non-deuterated methanol and ethanol, in order to show the effect of the solvent nuclear spins on the spin coherence times and on the modulation of the ESE signals measured after 2-pulse and 3-pulse sequences. Finally, cw-EPR measurements were also done on the same frozen solutions, albeit in the range 20–80 K due to signs of saturation at 6 K.

**Single-crystal X-ray diffraction**. Data for compounds [LaGd] and [GdLu] were collected at 100 K on a Bruker APEX II QUAZAR diffractometer equipped with a microfocus multilayer monochromator with MoKα radiation (λ = 0.71073 Å), respectively on a yellow lath and a yellow needle of dimensions $0.86 \times 0.18 \times 0.12$ and $0.40 \times 0.03 \times 0.03$ mm³. Data reduction and absorption corrections were performed with SAINT and SADABS[59]. Both structures were solved by intrinsic phasing with SHELXT[60] and refined by full-matrix least-squares on $F^2$ with SHELXL-2014[61]. In the case of [GdLu], remaining voids in the structure with no significant electron density peaks were analysed and taken into account with PLATON SQUEEZE[62] that recovered a total of 224 electrons per cell, over four equivalent voids of 193 Å³. These figures being reasonable for one diffuse pyridine solvent molecule per void, i.e., one per formula unit, this has been reflected in the reported formula. For both structures, the metal site composition was confirmed by refining the structure with the two homometallic situations as well as with the Ln sites inverted. These resulted in relatively poorer agreement factors and most importantly in unrealistic combinations of $U_{eq}$ values at the metal sites.

**Theory: universality test for molecular spin qudits**. A universal quantum processor must be able to implement any unitary operation within the computational states embedded in a Hilbert space. In the particular case discussed here, i.e., qudits simulating qubits: any transition between two molecular states must be driven by applying an external field. To show if a particular molecular qudit is universal (in this sense) or not, we write the spin Hamiltonian as,

$$\mathcal{H} = \mathcal{H}_s - g\mu_B \mathbf{B}_1(t)\mathbf{S}. \qquad (4)$$

Here $\vec{S}$ is the total spin operator and $\mathbf{B}_1(t)$ is a time-dependent external magnetic field that induces resonant transitions between different eigenstates of the spin Hamiltonian $\mathcal{H}_s$. These eigenstates form the computational basis. Since the dimension $d$ of the Hilbert spaces of all spin qudits considered in this work is $d = 2^N$, with $N = 3$ for [LaGd] and [GdLu], and $N = 6$ for [Gd₂], these eigenstates

may be denoted as either $|n\rangle$ (with $n = 1, ..., d$) or as $|00...0\rangle$ to $|11...1\rangle$. If all of these states are accessible from any other one, the qudit realizes a N-qubit processor.

One way to check universality is as follows. We use the fact that the energy spectrum has some nonlinearity, i.e., that the levels are not simply equidistant as those of a harmonic oscillator. This arises from the combination of single-ion anisotropy, the dissymmetry between the two Gd(III) coordination sites and the mutual interaction between the two Gd(III) spins. This means that we can address a transition between any two states, say $|n\rangle$ and $|m\rangle$, by making the frequency $\omega$ of the driving magnetic field $\mathbf{B}_1(t)$ resonant with the transition frequency $\omega_{nm} = (E_n - E_m)/\hbar$, provided that the matrix element $\langle n|\mathbf{B}_1(t)\mathbf{S}|m\rangle \neq 0$. If this happens, it is fair to say that we can implement any unitary operation of the form $e^{iH_{nm}t}$ with $H_{nm} = |n\rangle\langle m| + h.c.$ The allowed transitions define a set $L = \{H_{nm}, H_{n'm'}, ...\}$.

It is however expected that $L$ will not cover *all* the necessary transitions. The natural question is then which extra transitions can be implemented by concatenating the different elements of $L$. The formal answer[42] to this question is that the accessible transitions are those belonging to the Lie algebra $\mathcal{L}$ generated by $L$. This is a natural consequence of the Lie-Trotter formulas, i.e.

$$e^{i[H_{nm}, H_{n'm'}]t} = \lim_{C \to \infty} \left( e^{\frac{H_{nm}t}{\sqrt{C}}} e^{\frac{H_{n'm'}t}{\sqrt{C}}} e^{-\frac{H_{nm}t}{\sqrt{C}}} e^{-\frac{H_{n'm'}t}{\sqrt{C}}} \right)^C. \qquad (5)$$

The different commutators of the elements of $L$ are also elements of the corresponding Lie group, $\mathcal{L}$. In practice, to find the allowed transitions, we may compute the $H_{nm}$ by expressing $g\mu_B \mathbf{B}_1(t)\mathbf{S}$ in the basis of eigenstates of $\mathcal{H}_s$ and compute all possible commutators. By doing so, we can check if $\mathcal{L}$ covers the full Hilbert space. In a finite dimensional basis, this is easily computed since the non-zero commutators are of the form:

$$i[H_{nm}, H_{n'm'}] = i|m\rangle\langle m'| + h.c. \qquad (6)$$

We finally notice that the proof says nothing about how to perform operations. It only checks if they are possible.

**Universality of [GdLu] and [LaGd] three-qubit systems (or qu8its)**. These molecules have a spin Hamiltonian $\mathcal{H}_s$ given by Eq. (1). It is rather simple to prove that this Hamiltonian leads to universal operations. It turns out that for intermediate magnetic fields, say $\mu_0 H_z \sim 0.5 - 1$ T, the level spectrum consists of non-equidistant levels and that $\langle n|S_x|n+1\rangle \neq 0$ for all eigenstates $|n\rangle$ (see Supplementary Fig. 4). From the point of view of the universal operation, this is rather favourable since $i[H_{n\,n+1}, H_{n+1\,n+2}] = i|n\rangle\langle n+2| + h.c.$, and so on. Therefore, every transition can be performed by concatenating commutators.

**Universality of the [Gd₂] six-qubit system (or qu64it)**. Putting together several qudits, they must interact as a prerequisite to be universal. The [Gd₂] dimer is described by Eq. (2), which includes a weak, antiferromagnetic interaction, thus it fulfils this condition. Besides, from the previous discussion it follows that if some symmetry is shared by $\mathcal{H}_s$ and $\mathbf{B}_1(t)\mathbf{S}$, the system is presumably not universal. For example, a parity selection rule can impose that only transitions $\langle n|\mathbf{B}_1(t)\mathbf{S}|n+2\rangle \neq 0$. Then, this system becomes not universal: starting from an even labelled state $|n\rangle$, the odd states cannot be accessed, and viceversa. This condition might introduce difficulties in situations that are closer to that described by the spin Hamiltonian of [Gd₂]. The isotropic Heisenberg model, $\mathcal{H} = -J\mathbf{S}_1\mathbf{S}_2 - g\mu_B \mathbf{H}(\mathbf{S}_1 + \mathbf{S}_2)$ conserves the total spin $S$. Therefore, a spin dimer described by this model is definitely not universal for an external driving $g\mu_B\mathbf{B}_1(t)\mathbf{S}$, with $\mathbf{S} = \mathbf{S}_1 + \mathbf{S}_2$. Transitions between different total spin states are forbidden. Fortunately, in [Gd₂] the different anisotropy terms acting on each spin break this symmetry, besides providing a non-equidistant level spectrum. However, contrary to what happens in the single qudit case, the dc magnetic field $H$ cannot be much higher that $J$, $D$ and $E$. The reason is twofold. If $H$ is strong enough, the two Gd(III) ions are effectively decoupled. At the same time, the anisotropy terms become less important and the system is closer to an isotropic Heisenberg system. Our numerical calculations, whose results are shown in Fig. 7 and in Supplementary Fig. 5, confirm this. Figure 7b and the top panels of Supplementary Fig. 5 show maps of the Rabi frequencies for resonant transitions between any pair of states. Using transitions with Rabi frequencies larger than 0.2 MHz mT⁻¹, we construct contour plots (Fig. 7c and bottom panels of Supplementary Fig. 5) of those transitions that are feasible by concatenation of all possible commutation operators. The system is universal if the matrix built in this way spans all points (except the diagonal). Figure 7 shows the universal character of the dimer of two coupled Gd(III) ions in a not too strong magnetic dc-field. Then, Supplementary Fig. 5 shows how this property is broken down by either setting $J = 0$ or increasing the dc-external field. The conclusion is that [Gd₂] can, under the appropriate conditions, simulate either a six-qubit processor or two decoupled three-qubit systems.

## Data availability
Raw data sets relevant to this publication are freely available via the FATMOLS community at the ZENODO repository, using the link https://zenodo.org/communities/fatmols-fet-open-862893/about/. These raw data correspond to Figs. 2 to 9. The X-ray crystallographic coordinates for structures of [LaGd] and [GdLu] reported in this article

have been deposited at the Cambridge Crystallographic Data Centre (CCDC), under deposition numbers CCDC 1999444 and CCDC 1999445, respectively. The structure of [Gd$_2$] has been reported previously and can be found in CCDC 915331. These data can be obtained free of charge from The Cambridge Crystallographic Data Centre via https://summary.ccdc.cam.ac.uk/structure-summary-form. Crystallographic and refinement parameters for [LaGd] and [GdLu] are summarized in Supplementary Table 1. Selected details on hydrogen bonds are given in Supplementary Table 2. Bond lengths and angles are listed in Supplementary Table 3. The CIF files for [LaGd] and [GdLu] are also provided as Supplementary Data 1 and Supplementary Data 2, respectively.

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

## Acknowledgements

This work was supported by funds from the EU (ERC Starting Grant 258060 Func-MolQIP, COST Action 15128 MOLSPIN, QUANTERA project SUMO, FET-OPEN grant 862893 FATMOLS), the Spanish MICINN (grants CTQ2015–68370-P, CTQ2015–64486-R, RTI2018–096075-B-C21, PCI2018–093116, PGC2018–098630-B-I00, MAT2017–86826-R) and the Gobierno de Aragón (grants E09–17R-Q-MAD, E31_17R PLATON). G.A. thanks to the Generalitat de Catalunya for the prize ICREA Academia 2018.

## Author contributions

V.V., D.A., L.A.B. and G.A. designed and synthesized the molecular complexes. O.R. performed the heat capacity measurements, O.R. and F.L. performed the magnetic measurements and P.J.A., J.I.M. and O.R. performed the EPR measurements. F.L. modelled the heat capacity and magnetic data and P.J.A., J.I.M. and F.L. modelled the EPR results. D.Z. performed the quantum universality analysis. G.A. and F.L. conceived the idea and supervised the project. G.A., O.R. and F.L. wrote the paper with input from all co-authors.

## Competing interests

The authors declare no competing interests.
