## [Peer Review File · Communications Chemistry]

Reviewers' comments:

Reviewer #1 (Remarks to the Author):

The Paper by Luis and coworkers is a nice indepth structural and spectroscopic study of coherent spin control in well-designed dinuclear systems. Through careful synthetic design, they engage in "spin Hamiltonian engineering" to ensure that two Gd ions in a single molecule have different individual EPR responses and probe analogous dinuclear complexes of [LaGd] and [GdLu] to test the individual ions. Ultimately, I think the work is important, of interest to many others at the intersection of inorganic chemistry and quantum science, and worth publication. I do have some reservations about the presentation and key points, so I think it's worth revising and addressing some of the below comments prior to acceptance.

I think some of the wording in the title/abstract are aiming higher than the results of the data. I think it's a minor point, but one worth addressing. The authors demonstrate that each individual Gd ion has it's own set of spin Hamiltonian parameters, so in principle, sure, one could address each one to then perform a lot of the experiments the authors suggest. But the EPR spectrum, as shown, has really broad peaks, so it seems like some of the individual addressability is lost (this is especially true for the echo-detected spectrum).

The authors claim that partial scrambling occurs in solution but the process is very limited. I doubt that a small portion of the [GdGd] impurities would compromise the conclusions. But, the authors only say that the scrambling is very limited, and I think it would be more convincing to provide some addition data to tell us how they quantified the extent of scrambling.

Regarding the collection of T1, I'm curious as to why the Authors used a three p/2 pulse sequence for this? Typical T1 data are determined by Hahn-echo detected inversion recovery (π - variable delay – $\pi/2$ -t- π -t-echo) or saturation recovery experiments. As presented, I think the authors are quantifying T1 through a three-pulsed ESEEM experiment, which gives a relaxation time that is practically somewhere between T1 and T2, but not really T1. Can the authors elaborate on their rationale here? If they did a true inversion recovery, they might be able to see if the recovery curve can be fit with a mono v. biexponential function. A biexponential function might be able to quantitate heterogeneity of the samples (more relaxation times needed to fit = different species present), whereas monoexponential might indicate a largely homogeneous sample.

A significant value in the work is the ability to compare TM for the individual ions versus the coupled system and there define the mechanisms of decoherence. The authors have a really interesting claim (the factors that contribute to TM for "Gd1" in [GdGd] also impact "Gd2", presumably through the exchange coupling. I think that's a fascinating idea (and worth exploring synthetically in a future study), but I'm not sure the authors have the data to show that, as you'd have to be able to individually map T2 for one ion to the other, and they only have one complex here. The authors also say that the change in TM as a function of field is because of the competition between spin-spin coupling and the zeeman interaction. I'm wondering if there is a way that they can rule out orientation effects from causing it, as is well-documented in mononuclear species, but not so much (to the best of my knowledge) for dinuclear compounds. Some simulations with Easyspin should be able to rule this out.

In terms of a future work, I think it's clear from this that an important next challenge is simply figuring out how to experimentally control the EPR linewidth and ensure addressability. This is a very nice study in spin Hamiltonian engineering, but I think the linewidth challenge is difficult, and I think the authors should consider or mention that as part of their outlook.

Reviewer #2 (Remarks to the Author):

The manuscript, A dissymmetric [Gd₂] coordination molecular dimer hosting six addressable spin qubits, by F. Luis et. al. describes the magnetic characterization of a set of Gd complexes and frames them as potential as qubits and qudits for applications in quantum information science. The molecules were designed in order to leverage the high spin nature of Gd(III) and move beyond molecular qubits into a higher order molecular qudit. This is an interesting paper and with some revisions I believe it would be appropriate for Communications Chemistry.

The Authors should address the following points:

1. Minor point, on line 161-162 it would be helpful to initially define L.
2. The x axis label for the EPR spectra, figures 2,5,7,8, should be labelled as Magnetic Field rather than μOH . This is in order to bring it to the standard for reporting EPR spectra and to reduce confusion in the broader audience.
3. Kelvin is not exactly a standard unit to report energy nor EPR parameters. The interaction energies should be changed to MHz/GHz, this would bring things in line with what is standard in the field. Also, this change would help to reduce confusion and make the manuscript more approachable to the broader audience.
4. On figure 2E, labels on the energy levels would be helpful. It is unclear from the figure caption and the text which basis is being plotted, is it $m_s = \pm 1/2, 3/2, 5/2, 7/2$? Or some other linear combination of spin states? This is potentially important in order to draw corollaries with figure 7A and their viabilities as single qubits/qudits at zero field.
5. With the spin Hamiltonian, line 191, in molecular species, D and E are commonly referred to as the zero field splitting parameters, rather than anisotropy constants. The reference to ZFS on line 206 doesn't help. In addition, g is the electron spin g-factor, which is a dimensionless quantity, the gyromagnetic factor has μB folded into it, so either the definition or Hamiltonian should be changed.
6. The sign E was reported for [LaGd], while the absolute of E is reported for [GdLu] why was the sign of E not reported? What is the significance of the sign of those interactions? Given that E/D is effectively 1/3 for both molecules, so a fully rhombic ZFS tensor, what does that mean with respect to the structure? Since the crystal structures is reported, is it possible to estimate, or calculate the orientation of the ZFS relative to the molecular axis.
7. As stated on lines 256-7 the intramolecular dipole-dipole interaction is potentially quite large, on the same order of magnitude as the intermolecular ZFS, as such this interaction really should be included in the EPR simulations of Figure 5 and not just brushed aside.
8. Minor point, the difference between X-band and Q-band is more than just the resonator. Line 262 should probably read: results measured at X-band and Q-band.
9. In equation 3, γ_0 should be unnecessary if a 2 step or 4 step phase cycle was utilized. The experimental details regarding the pulse measurements are sparse, and generally lacking from the experiments section, what's the pulse duration? TWT or solid-state pulse amplifier? Was a phase cycle used? Which resonators were used?
10. Further on equations 3, strictly speaking it should be $\{1 - k \sin \dots\}$. ESEEM manifests itself as a modulation which decreases the echo envelope.
11. Line 376 sensible -> sensitive
12. The ESEEM frequency observed in the 2P and 3P is likely from the solvent deuterium. If the authors disagree and wish to stick to their originally proposed source of modulation several things should be addressed. In the manuscript it is suggested to come from ¹⁵N, was there isotopic labelling? ¹⁵N is only 0.368% abundant. Further, an explanation of why the modulation frequency is at twice the Larmor frequency would be necessary.
13. It is unclear on the point of including Rabi oscillations/transient nutation spectra. The presence of a spin echo as demonstrated in the 2-pulse and 3-pulse ESE measurements is evidence of coherence in the spin ensemble. In addition, any spin system that has relaxation times sufficient to provide a spin echo will also give a transient nutation. And this reviewer would argue that the nutation experiment as presented is not actually a good demonstration of coherent control in this particular spin system. As mentioned earlier in the manuscript (universality) there is a rather large range of nutation frequencies present in the spectrum, and many of them are simultaneously excited with the microwave pulse. Without actually providing any selectivity in the excitation it is

hard to believe the claim of coherent control. Also, this lack of selectivity is likely the main contributor to the damped oscillations. Perhaps a better demonstration of the nutation frequency, and experimentally confirming the universality calculations, would be the PEANUT experiment(<https://doi.org/10.1006/jmre.1997.1285>) and correlate the broad frequency envelope to magnetic field.

14. In the discussion, several of the future challenges are present, what about initialization of these systems?

15. Minor point, in the SI, figure S16, a note about the resolution is provided, why did the authors not zerofill the spectrum prior to Fourier transforming as a way to boost the frequency resolution. The resolution of individual features is dictated by how fast they damp.

Reviewer #3 (Remarks to the Author):

The manuscript by Fernando Luis et al. presents the design, synthesis and theoretical characterization of dissymmetric molecular dimmers hosting either one or two Gd ions. The main claim of the manuscript is that the spin degrees of freedom can be coherently controlled and can give rise to a multidimensional (up to 64 levels) Hilbert space. This claim is supported by a significant set of measurements, and by simulations. While most experimental claims are supported by measurements on solutions and powders, unfortunately the spectroscopy of the 6 qubit register and its control are only demonstrated in simulations.

While molecular spins have some interesting potential for quantum registers, the current level of quantum coherence, including the values demonstrated in the manuscript need to improve by many orders of magnitude. As the authors mention, better stability can presumably be expected for diluted solids as well as for isolated molecules, possibly integrated into hybrid devices.

The text is well structured, including technical details and a relevant selection of bibliographic references. The claims are generally supported by experimental data. One overarching remark I have is that the text feels too strongly focused on quantum information processing, while the coherence of the ensembles barely allows pulses. I suggest toning down some of the statements regarding quantum information processing, such as the "universality", which is only supported by simulations.

In view of the quality of the work, its originality, and the possible appeal to neighboring fields, such as the superconducting devices community, I support the publication of the manuscript.

I would have one detailed remark: several plots in Fig. 5, 8 and 9 do not have a Y-axis. This can be confusing for a reader not familiar with the community jargon.

Responses to reviewers' comments

Reviewer #1:

We thank the referee for considering our work worth publishing and for his/her insightful comments and. Below, we give responses to all of them.

Comment 1: I think some of the wording in the title/abstract are aiming higher than the results of the data. I think it's a minor point, but one worth addressing. The authors demonstrate that each individual Gd ion has it's own set of spin Hamiltonian parameters, so in principle, sure, one could address each one to then perform a lot of the experiments the authors suggest. But the EPR spectrum, as shown, has really broad peaks, so it seems like some of the individual addressability is lost (this is especially true for the echo-detected spectrum).

Response 1: In our opinion, what our work brings is, as the referee correctly points out, the synthesis of a molecular structure able to host two Gd ions, each of them with a different magnetic anisotropy, sufficiently strong to make them magnetically distinguishable but also sufficiently weak as to keep resonance transition frequencies between their 8 different levels experimentally accessible, and with a coupling between them that fulfils the same criteria. Besides, we have shown that the ensuing spin Hamiltonian, backed with solid experimental evidence, is compatible with universal operations. And that spin coherence times are reasonably long, at least if one compares them with the attainable Rabi frequencies. Even if a complete implementation of quantum gates is precluded by the intrinsic limitations of experiments performed on frozen solutions, which are necessary for getting measurable signals and sufficiently long coherence times, we still believe that this work shows a synthetic path towards the chemical engineering of well defined molecular structures that can embed non-trivial, potentially very useful, quantum functionalities.

In the revised version, we have rewritten several sections of the manuscript, namely the abstract, the section on universality, the discussion on the nutation experiments and the conclusions, in order to express more clearly what the main results are, but also their limitations and the challenges lying ahead.

C2: The authors claim that partial scrambling occurs in solution but the process is very limited. I doubt that a small portion of the [GdGd] impurities would compromise the conclusions. But, the authors only say that the scrambling is very limited, and I think it would be more convincing to provide some addition data to tell us how they quantified the extent of scrambling.

R2: We agree with the referee that the issue of the potential [GdGd] impurity deserves a deeper discussion, since one could think that this situation may affect the conclusions while, as we argue next, they do not. We cannot provide data to quantify the extent of the scrambling for various reasons, as explained here and in the main text of the revised manuscript. First, because ESI-MS does not allow to quantify the relative amounts of the species in solution. And most importantly, because the possible mixture [GdLu]/[GdGd] examined is that of the fraction in solution with the terminal ligands (H₂O, pyridine and NO₃⁻) dissociated. This fraction is presumably very minor since most complexes are expected to conserve their terminal ligands, but the latter species are not detected by MS. The selectivity

of the latter (thus, of the majority) is close to that in the solid (since the molecular moieties are much more alike), while the selectivity of the species with no terminal ligands is lower, as has been shown by previous work by some of the authors (references [16] and [17] of the manuscript). In these, extensive DFT calculations were applied to evaluate the reduced selectivity of the dinuclear complex with no terminal ligands for a number of pairs of lanthanide ions with varying Δr_i . In the case of [GdLu], the Δr_i remains large, and the extent of scrambling of the fragment with dissociated terminal ligands is not expected to be significant. While we cannot discard the presence of some [GdGd] impurities in solutions of [GdLu], for the above reasons we are confident that this remains very limited. This is supported by the following: *i)* the ei-EPR spectrum of [GdLu] obtained using a 3-pulse detection do not show any shoulder in the field range at which the [GdGd] exhibits a maximum (Fig. S8); *ii)* The T2 value measured for [GdLu] is longer than that of [LaGd], which in turn is longer than the one measured for [GdGd]. This observation, in addition, confirms that the conclusions of this work would not be affected by the presence of a small [GdGd] impurity in the [GdLu] solution.

C3: Regarding the collection of T1, I'm curious as to why the Authors used a three p/2 pulse sequence for this? Typical T1 data are determined by Hahn-echo detected inversion recovery (pi- variable delay – pi/2-t-pi-t-echo) or saturation recovery experiments. As presented, I think the authors are quantifying T1 through a three-pulsed ESEEM experiment, which gives a relaxation time that is practically somewhere between T1 and T2, but not really T1. Can the authors elaborate on their rationale here? If they did a true inversion recovery, they might be able to see if the recovery curve can be fit with a mono v. biexponential function. A biexponential function might be able to quantitate heterogeneity of the samples (more relaxation times needed to fit = different species present), whereas monoexponential might indicate a largely homogeneous sample.

R3: We agree with the referee. Inversion recovery measurements provide a more direct information on the relaxation of the longitudinal spin component, thus of T1, and are free from the influence of spin coherences and their modulation by hyperfine couplings. We have performed such experiments on solutions of the three samples. The results show similar T1 values as those derived from the three-pulse experiments. Therefore, these new measurements do not affect the main conclusions. We have replaced the relevant figures in the supplementary material and revised the text accordingly.

C4: A significant value in the work is the ability to compare TM for the individual ions versus the coupled system and there define the mechanisms of decoherence. The authors have a really interesting claim (the factors that contribute to TM for "Gd1" in [GdGd] also impact "Gd2", presumably through the exchange coupling. I think that's a fascinating idea (and worth exploring synthetically in a future study), but I'm not sure the authors have the data to show that, as you'd have to be able to individually map T2 for one ion to the other, and they only have one complex here. The authors also say that the change in TM as a function of field is because of the competition between spin-spin coupling and the zeeman interaction. I'm wondering if there is a way that they can rule out orientation effects from causing it, as well-documented in mononuclear species, but not so much (to the best of my knowledge) for dinuclear compounds. Some simulations with Easyspin should be able to rule this out.

R4: While not a central issue to our work, we agree that the comparison of TM measured in the monomers ([GdLu] and [LaGd]) and the [GdGd] is fascinating, as it might point out to

decoherence mechanisms operating in entangled spin states that are different from those limiting coherence of each of the constituents. Studying this effect in detail requires synthesizing a family of complexes, hosting single ions and pairs of them. This study, made possible by the nice features of the present synthetic system, is currently in progress. Yet, we believe it is worth mentioning this point here, even if as a possible explanation of the observations.

Concerning the experimental evidence, we actually derive our suggestion from the study of three complexes, which allow measuring TM of each of the Gd ions in their respective coordination sites and that of the two coupled ions (in the same sites) under the same experimental conditions. We find a consistently lower TM for the latter case, and quite similar values for the monomers which we tentatively ascribe to the fact that the states are not simple products of states of the two Gd(III) ions (thus each transition is not just rotating one in the field created by the other).

Although the field dependence might partly be associated with a gradual change in the number of transitions being explored by the pulses (which become more selective to certain molecular orientations near the edges of the spectrum), this would apply similarly to the samples of both monomers and of the dimer as well. By contrast, the differences between the monomers and the dimer TMs are reduced by increasing H. In addition, we have found that other physical quantities measured on [GdGd] (chiefly the heat capacity) also tend to gradually approach the superposition of those measured on [LaGd] and [GdLu] with increasing H. Taking all into consideration, we still believe that our interpretation, in terms of a stronger sensitivity of [GdGd] states to decoherence, makes sense. In any case, in the revised version we give it as a tentative explanation, waiting for a more complete study of this question.

C5: In terms of a future work, I think it's clear from this that an important next challenge is simply figuring out how to experimentally control the EPR linewidth and ensure addressability. This is a very nice study in spin Hamiltonian engineering, but I think the linewidth challenge is difficult, and I think the authors should consider or mention that as part of their outlook.

R5: We agree, of course. In our revision of the section devoted to conclusions and outlook, we have tried to list the limitations of our experiments with regards implementation and discussed possible ways to circumvent this problem. Although some of the possible strategies involve novel chemical routes to produce diluted crystals of these dimers in diamagnetic hosts, which lies within the realm of chemistry and materials science but is also very challenging in view of the possible ionic inter-exchange between molecules in solution, progress in this direction will probably also come along with the development of new tools to control and detect the response of individual molecules, something that falls outside the scope of the present work.

Reviewer #2 (Remarks to the Author):

We thank the referee for considering our work appropriate for Communications Chemistry and for his/her insightful comments and. Below, we give responses to all of them.

Comment 1: Minor point, on line 161-162 it would be helpful to initially define L.

Response 1: We have done so in the revised version.

C2: The x axis label for the EPR spectra, figures 2,5,7,8, should be labelled as Magnetic Field rather than $\mu\text{O}H$. This is in order to bring it to the standard for reporting EPR spectra and to reduce confusion in the broader audience.

R2: We have changed the labelling of the magnetic field in all figures of the main text and the SI.

C3: Kelvin is not exactly a standard unit to report energy nor EPR parameters. The interaction energies should be changed to MHz/GHz, this would bring things in line with what is standard in the field. Also, this change would help to reduce confusion and make the manuscript more approachable to the broader audience.

R3: While Kelvin is a rather frequent unit of energy in connection with studies of Magnetism and of the thermal properties of materials, it is certainly less so when dealing with spectroscopic measurements. Our idea to combine different units was to allow the readers to compare the energy splittings in these molecules with the energy of the EPR cavity photons as well as with the temperature scales in heat capacity and susceptibility measurements. However, we understand this could also bring some confusion. Therefore, since using just the same units throughout the manuscript seems like a very sensible suggestion, we have adopted it in our revision.

C4: On figure 2E, labels on the energy levels would be helpful. It is unclear from the figure caption and the text which basis is being plotted, is it $m_s = \pm 1/2, 3/2, 5/2, 7/2$? Or some other linear combination of spin states? This is potentially important in order to draw corollaries with figure 7A and their viabilities as single qubits/qudits at zero field.

R4: The revised Figure 4 now includes information on the wave functions of the different levels at zero field. They are certainly not pure spin projections.

C5: With the spin Hamiltonian, line 191, in molecular species, D and E are commonly referred to as the zero field splitting parameters, rather than anisotropy constants. The reference to ZFS on line 206 doesn't help. In addition, g is the electron spin g-factor, which is a dimensionless quantity, the gyromagnetic factor has μ_B folded into it, so either the definition or Hamiltonian should be changed.

R5: We fully agree. When referring to the gyromagnetic ratio we actually meant the Landee factor, but we have adopted the simpler and more accurate nomenclature suggested by the referee.

C6: The sign E was reported for [LaGd], while the absolute of E is reported for [GdLu] why was the sign of E not reported? What is the significance of the sign of those interactions? Given that E/D is effectively 1/3 for both molecules, so a fully rhombic ZFS tensor, what does that mean with respect to the structure? Since the crystal structures is reported, is it possible to estimate, or calculate the orientation of the ZFS relative to the molecular axis.

R6: This was a mistake indeed. The fit of cw-EPR spectra gives only information on the magnitude, but not the sign of the E parameter. The local coordination sites for Gd in these molecular structures have a very low symmetry (see Fig. S2), which probably leads to the

large, close to maximum, orthorhombicity we observe. A first-principles determination of the anisotropy parameters and of the orientations of the two anisotropy tensors is a quite hopeless task in this particular case for two main reasons. The first is that, contrary to what happens with all other lanthanides, the anisotropy of Gd(III) arises from a higher order perturbation involving the mixing of the ground state multiplet (with $S=7/2$ and $L=0$, thus no intrinsic anisotropy) with excited states via the spin-orbit interaction. The second, connected with this, is that the energy splittings are very small, thus quite demanding for ab-initio methods. For these reasons, we have preferred to rely of the fit of experimental quantities and adopted a simplified version of the spin Hamiltonian (with collinear anisotropy tensors) that, while probably not being the most general is still able to account for the experiments. As we emphasize in the manuscript (more so after revision), the parameters in the [Gd₂] molecule have to be seen as effective descriptions of the low-energy level scheme. It is worth mentioning here that any deviation from non-collinearity (and any anisotropy in the spin-spin interactions) would remove selection rules for resonant transitions, thus widen the range of parameters (intensity and orientation of magnetic field) for which the [Gd₂] system allows universal operations. Our approach must therefore be regarded as conservative in this point.

C7: As stated on lines 256-7 the intramolecular dipole-dipole interaction is potentially quite large, on the same order of magnitude as the intermolecular ZFS, as such this interaction really should be included in the EPR simulations of Figure 5 and not just brushed aside.

R7: The arguments given in R6 above justify also the difficulty in calculating the orientations of the two anisotropy tensors (i.e. of the three principal axes for each coordination site) using ab initio or related methods. These would be required to properly derive the dipolar interaction between the two ions. For this reason, in the spin Hamiltonian of [Gd₂] (Eq (2)), we introduce the intramolecular coupling between the two Gd spins by the simplest isotropic exchange term. Again, this must be regarded as a way to effectively parameterize such interactions. It is worth noticing that we did not brush these coupling aside. The simulations of EPR, heat capacity and magnetic susceptibility data of [Gd₂] are all done using the same spin Hamiltonian with a finite coupling constant J . Also, any deviation from the pure isotropic model, which is likely to be the case when dealing with dipolar interactions, helps universal operation as described in the methods section devoted to this issue.

C8: Minor point, the difference between X-band and Q-band is more than just the resonator. Line 262 should probably read: results measured at X-band and Q-band.

R8: We agree and we have made the change suggested.

C9: In equation 3, y_0 should be unnecessary if a 2 step or 4 step phase cycle was utilized. The experimental details regarding the pulse measurements are sparse, and generally lacking from the experiments section, what's the pulse duration? TWT or solid-state pulse amplifier? Was a phase cycle used? Which resonators were used?

R9: A background y_0 , independent of magnetic field, is added to Eq. (3) because the zero of the electronic output signal of the spectrometer is floating. Concerning the experimental details: the typical width of the $\pi/2$ and π pulses was 16 ns and 32 ns, respectively. In order to avoid unwanted echoes, a 2 step (4 step) phase cycle was used in the 2-(3-) pulse experiment. The high power microwave excitation was obtained by using a TWT amplifier. A

dielectric low Q cavity from Bruker was used as resonator. All these details have been added to the revised manuscript and to the SI.

C10: Further on equations 3, strictly speaking it should be $\{1 - k \sin \dots\}$. ESEEM manifests itself as a modulation which decreases the echo envelope.

R10: Formally speaking, we agree with the referee. Notice, however, that $\sin(x+\pi/2) = \cos(x)$ and that $\sin(x+\pi) = -\sin(x)$, from which it follows that $-k\sin(2\pi\nu\tau+\phi+3\pi/2) = k\cos(2\pi\nu\tau+\phi)$, thus both expressions bear the same physical meaning. The extra decaying factor $\exp(-\lambda\tau)$ accounts for the distribution, possibly inhomogeneous, of the modulation frequency whose average value equals the nuclear Larmor frequency (or, depending on the nucleus involved, twice the Larmor frequency).

C11: Line 376 sensible -> sensitive.

R11: We have also corrected this error.

C12: The ESEEM frequency observed in the 2P and 3P is likely from the solvent deuterium. If the authors disagree and wish to stick to their originally proposed source of modulation several things should be addressed. In the manuscript it is suggested to come from ^{15}N , was there isotopic labelling? ^{15}N is only 0.368% abundant. Further, an explanation of why the modulation frequency is at twice the Larmor frequency would be necessary.

R12: Mentioning ^{15}N was a mistake that unfortunately propagated via “copy and paste” to several places. We are very sorry for this. The frequency of the given signal agrees with the “double Larmor” transition of the most abundant isotope of Nitrogen ^{14}N , as was correctly stated in some of the SI figure captions. This is typically the dominant modulating signal when this element is present, because of its weak dependence on orientation, while the “single Larmor” broadens in a randomly oriented sample and therefore often effectively disappears. We have changed the text accordingly.

C13. It is unclear on the point of including Rabi oscillations/transient nutation spectra. The presence of a spin echo as demonstrated in the 2-pulse and 3-pulse ESE measurements is evidence of coherence in the spin ensemble. In addition, any spin system that has relaxation times sufficient to provide a spin echo will also give a transient nutation. And this reviewer would argue that the nutation experiment as presented is not actually a good demonstration of coherent control in this particular spin system. As mentioned earlier in the manuscript(universality) there is a rather large range of nutation frequencies present in the spectrum, and many of them are simultaneously excited with the microwave pulse. Without actually providing any selectivity in the excitation it is hard to believe the claim of coherent control. Also, this lack of selectivity is likely the main contributor to the damped oscillations. Perhaps a better demonstration of the nutation frequency, and experimentally confirming the universality calculations, would be the PEANUT experiment(<https://doi.org/10.1006/jmre.1997.1285>) and correlate the broad frequency envelope to magnetic field.

R13: We agree in that the results of nutation experiments are not a better (nor even comparable perhaps) demonstration of spin coherence than the ESE measurements. The reason is obviously that we are dealing with a random orientation of molecules in a frozen

solution, and that each molecule already possesses several allowed transitions. Then, the unavoidable distribution of Rabi oscillations excited by a single pulse leads to a rapid destructive interference. Yet, these experiments provide valuable information on the Rabi frequencies and their distribution, which can be compared (and that agree quite well) with those derived from the spin Hamiltonian and that have been used in the discussion about universal quantum operation. In addition, these experiments illustrate the limitations that this field of research is facing when trying to come closer to actual implementations in molecular spin qubits.

In the revised version, we have reduced the section devoted to describe these results, explained more clearly the origin of the strong damping and toned down their relevance regarding coherent control. Also, in our revision of the concluding section, we discuss the limitations that experiments on frozen solutions, which still provide one of the few possibilities to study spin coherence in these systems, are facing and suggest possible ways to progress (even if challenging).

C14. In the discussion, several of the future challenges are present, what about initialization of these systems?

R14: We thank the referee for pointing out this important question. One of the advantages of electron spin qubits over nuclear spin qubits is that the energy splittings involved are often larger, thus affording the possibility of initializing the spin state by cooling to experimentally accessible temperatures. For instance, a nearly complete spin polarization of [Gd²⁺] can be achieved by cooling down to temperatures below 100 mK under magnetic fields of the order of 0.5 T. This possibility is now explicitly mentioned near the end of the manuscript.

C15. Minor point, in the SI, figure S16, a note about the resolution is provided, why did the authors not zerofill the spectrum prior to Fourier transforming as a way to boost the frequency resolution. The resolution of individual features is dictated by how fast they damp.

R15: Figures S15 and S16 provide data measured on the same [GdLu] sample. The peaks associated with the coupling to ¹H and ¹⁴N nuclear spins are well resolved in the spectra shown in Fig. S15. Those shown in Fig. S16 provide extra information on the distributions of Rabi frequencies at some other magnetic field values. Besides, the modulation signal has not yet vanished for the longest times measured. The zerofill technique would then introduce a fictitious width. For these reasons, we preferred to keep the data in Fig. S16 as originally measured and simply state their limitations in terms of frequency resolution.

Reviewer #3 (Remarks to the Author):

We thank the referee for supporting the publication of our work in Communications Chemistry and for his/her insightful comments and. Below, we give responses to all of them.

Comment 1: The main claim of the manuscript is that the spin degrees of freedom can be coherently controlled and can give rise to a multidimensional (up to 64 levels) Hilbert space. This claim is supported by a significant set of measurements, and by simulations. While most experimental claims are supported by measurements on solutions and powders, unfortunately the spectroscopy of the 6 qubit register and its control are only demonstrated in simulations.

Response 1: The experiments are indeed performed on randomly oriented samples. Yet, we show that Eq. (2) provides a reasonably good account of all quantities (magnetic susceptibility, heat capacity but also magnetic spectroscopy) measured on [Gd₂]. It is, however, true, that these experiments have intrinsic limitations with regards the actual implementation of quantum gates, nor to say algorithms. In the revised version, and in the responses to referees 1 and 2 above, we discuss these limitations and suggest possible ways to progress towards such implementations.

C2: The text is well structured, including technical details and a relevant selection of bibliographic references. The claims are generally supported by experimental data. One overarching remark I have is that the text feels too strongly focused on quantum information processing, while the coherence of the ensembles barely allows pulses. I suggest toning down some of the statements regarding quantum information processing, such as the “universality”, which is only supported by simulations.

R2: As discussed in previous responses to similar comments by referees 1 and 2, we have made our best to properly describe what the main achievements of this work are, and avoid any empty boasting. We critically discuss limitations of the present experimental approaches, which use solid frozen solutions in order to combine a measurably signal and long enough TM.

However, it is worth noticing that the spin coherence in these systems is quite competitive, at least comparable, with what is found for other lanthanide spin qubits. The fact that Rabi oscillations show very fast damping arises from the interference of the different oscillations that are excited by the same pulse (a result of the multi-level nature of each molecule and their random orientation) rather than from the values of TM as compare to the Rabi frequencies.

The demonstration of universality follows from the structure of the spin Hamiltonians (Equations (1) and (2)), which have been derived from the fit of different experimental quantities. It is also based on the Rabi frequencies calculated with these equations, which agree with those determined experimentally. In our opinion, this discussion makes sense if only because it has received but little attention in connection with research on molecular spin qubits (an exception is reference [29], which reports work done by some of the authors). Fulfilling this condition is crucial in order to assign to any multilevel system the ability to encode multiple qubits. And it is furthermore far from evident, as it depends on the symmetry of the spin Hamiltonian (in our case, that the two Gd spins have some anisotropy and that they are coupled) and on experimental conditions (e.g. strength and orientation of the external magnetic field). Examples of what can go wrong are shown in Fig. S5. For this reason, we still believe that it makes sense to retain this discussion. The revised version describes it more critically and more clearly, we hope.

C3: I would have one detailed remark: several plots in Fig. 5, 8 and 9 do not have a Y-axis. This can be confusing for a reader not familiar with the community jargon.

R3: We have followed the suggestion by the referee and added Y axes, labels and titles to these figures.

Lists of changes

Main text:

- The main text has been thoroughly revised. Major changes apply to the abstract and to sections dealing with the universality condition, the spin coherence and nutation experiments, the conclusions and the spin resonance methods. All these changes are marked with a yellow background.

Figures:

- We have replaced former Figs. 2, 4, 5, 6, 7 and 8 with new ones, in order to either include new information suggested by the reviewers or to modify the notation and labelling, also following their instructions. The bold face titles of these figures have also a yellow background to indicate that they have been modified.
- The captions of all these figures, plus that of Fig. 9, have also been revised accordingly.

References:

- Three references [42-44] have been reordered, following the text editions, and a new reference [54] to a recently published manuscript dealing with the development of spin resonance techniques able to detect small spin ensembles, has been added.

Supplementary information

- We have replaced the three panels of Fig. S11, which showed results of ESE decays recorded in three-pulse experiments, with new ones, showing the results of inversion recovery experiments.
- We have replaced former Fig. S13, which reported the spin-relaxation times and decay amplitudes derived from three pulse experiments, with the same data obtained from the new inversion recovery experiments.
- The captions of all these figures have also been revised accordingly.
- We have replaced the symbol $\mu_{0}H$ with magnetic field throughout the document, including figure captions and labelling.

As with the main text, changes to the SI are also indicated with a yellow background

Reviewers' comments:

Reviewer #1 (Remarks to the Author):

The authors have, in my opinion, addressed the major points that I brought up in my prior review, and I thank them for addressing my concerns. For this reason, I think the paper is good for publication. There are some minor typographical errors (e.g. "tipically" where "typically" should be), so the authors should carefully look over the manuscript to check for and fix the small errors prior to acceptance, however.

Reviewer #2 (Remarks to the Author):

The authors have adequately addressed most of my concerns and the manuscript is in a far better position. And I support publication as long as the authors address one final concern.

I mentioned it in my first round of reviews, the modulation observed in the 2-pulse data, the transient nutation, and the 3-pulse data was attributed to nitrogen, I disagree with that assignment. Those modulations should in fact be attributed to the solvent deuterons.

In two pulse ESEEM sum and difference frequencies can occur, which would provide the possibility of a line at it twice the Larmor frequency. However, in three pulse ESEEM which would encompass the effect observed in the nutation and the three pulse measurements you don't observe modulation at sum and difference frequencies, only at the transition frequencies. So, there is no reason to expect that the ESEEM from pyridine would manifest itself as only at twice the Larmor frequency in both two pulse and three pulse experiments, while also neglecting the contribution to the ESEEM from the solvent deuterons, which due to the sheer number of them typically dominate any modulation.

If this is not convincing I would suggest to the authors that they should probably try running 2- or 3- pulse ESEEM in a protonated or spin free solvent. Unless I am terribly mistaken, the low frequency peak should disappear and get replaced with a peak at the hydrogen Larmor frequency. Though in the three pulse ESEEM you could set your tau such that you burn a hole in the spectrum at that frequency and never observe it... If I am mistaken, it might be helpful to convince skeptics like me to include that result in the SI.

Reviewer #3 (Remarks to the Author):

I am satisfied with the changes made by the authors following my remarks. I support the publication of the manuscript.

Responses to reviewers' comments

Reviewer #1:

We thank the reviewer for recommending the publication of our manuscript. In the revised version, we have corrected the typographical error that he pointed out and carefully proof-read the text.

Reviewer #2:

We thank the reviewer for supporting the publication of our manuscript and for his very important comment on the modulation of the ESE signals. Our response follows below.

Comment 1: I mentioned it in my first round of reviews, the modulation observed in the 2-pulse data, the transient nutation, and the 3-pulse data was attributed to nitrogen, I disagree with that assignment. Those modulations should in fact be attributed to the solvent deuterons.

In two pulse ESEEM sum and difference frequencies can occur, which would provide the possibility of a line at it twice the Larmor frequency. However, in three pulse ESEEM which would encompass the effect observed in the nutation and the three pulse measurements you don't observe modulation at sum and difference frequencies, only at the transition frequencies. So, there is no reason to expect that the ESEEM from pyridine would manifest itself as only at twice the Larmor frequency in both two pulse and three pulse experiments, while also neglecting the contribution to the ESEEM from the solvent deuterons, which due to the sheer number of them typically dominate any modulation. If this is not convincing I would suggest to the authors that they should probably try running 2- or 3- pulse ESEEM in a protonated or spin free solvent. Unless I am terribly mistaken, the low frequency peak should disappear and get replaced with a peak at the hydrogen Larmor frequency. Though in the three pulse ESEEM you could set your tau such that you burn a hole in the spectrum at that frequency and never observe it... If I am mistaken, it might be helpful to convince skeptics like me to include that result in the SI.

Response 1: Following his/her suggestion, we undertook experiments on a fresh sample of [GdLu] dissolved in a non-deuterated ethanol-methanol mixture. The concentration and solvent composition, as well as temperature and magnetic fields were the same as those used for the previous experiments, with the sole difference in the isotopic labelling of the solvent. In addition, we have repeated the experiments on a deuterated solution, to ensure both were carried out under identical conditions. We found that the low-frequency modulation disappears from the ESE signal obtained by 2-pulse and 3-pulse experiments and was replaced with the modulation characteristic of the coupling to protons. An illustrative example has been added to the SI (new Fig. S6A). These results confirm the reviewer's view on the origin of this modulation and show that the couplings to ¹⁴N nuclei are not very relevant. In conclusion, we can safely attribute the modulation to the hyperfine interactions with remote solvent deuterons, exactly as the referee suggested. The change of this assignment does not modify the main conclusions of our manuscript. Needless to say, we have properly corrected this issue both in the manuscript and in the SI. We sincerely thank the referee for pointing up this question that will prevent the publication and propagation of inaccurate conclusions.

Reviewer #3 (Remarks to the Author):

We thank the referee for supporting the publication of our work in Communications Chemistry.

Lists of changes

Main text:

- The main text, chiefly the sections dealing with pulse EPR experiments, has been edited in order to comply with the suggestion made by the second referee, and thoroughly proofread to eliminate typos.
- The methods section refers now to the additional pulse-EPR experiments on non-deuterated solutions and describes the experimental conditions that applied to these experiments.

All these changes are marked with a yellow background.

Figures:

- We have replaced former Fig. 9 with a new one with the correct labelling for the ESE modulation peaks.
- The caption of this figure has also been revised accordingly.
- We provide all figures as separate files (a 300 dpi resolution PNG file for Fig. 1 and EPS files for the rest), as required for publication.

Supplementary information

- We have added a new Fig. S6A that shows the decay of ESE signals following from 2-pulse and 3-pulse experiments performed on solutions of [GdLu] in deuterated and non-deuterated solvents.
- We have replaced former Figs. S12, S14 and S16 with new ones, which include the correct labelling for the additional oscillations arising from the coupling of Gd spins to solvent deuterons.
- The captions of all these figures have also been revised accordingly.

As with the main text, changes to the SI are also indicated with a yellow background.

Style and formatting

- We have shortened the abstract and reformatted the text, equations, symbols and figures in order to comply with the style and formatting guidelines of communication journals.

REVIEWERS' COMMENTS:

Reviewer #2 (Remarks to the Author):

Looks good, all of my concerns have been addressed and I recommend going forward with publication.